# Molecular epidemiological characteristics of dengue virus carried by 34 patients in Guangzhou in 2018

Feng Liao[1][☯], Huini Chen[2][☯], Jieliang Xie[3], Shaofeng Zhan[4], Pan Pan[5], Zizhao Lao[6], Yaohua Fan[4], Lupin Lin[7], Yanni Lai[4], Shuangfeng Lin[4], Jianguo Wu[2], Xiaohong Liu[1], Geng Li [1]*

**1** Laboratory Animal Center, Guangzhou University of Chinese Medicine, Guangzhou, China, **2** Institute of Medical Microbiology, Jinan University, Guangzhou, China, **3** School of Pharmaceutical Sciences, Guangzhou University of Chinese Medicine, Guangzhou, China, **4** The First Affiliated Hospital of Guangzhou University of Chinese Medicine, Guangzhou, China, **5** College of Life Sciences, WuHan university, Wuhan, China, **6** Mathematical Engineering Academy of Chinese Medicine, Guangzhou University of Chinese Medicine, Guangzhou, China, **7** Guangzhou eighth People's Hospital, Guangzhou Medical University, Guangzhou, China

☯ These authors contributed equally to this work.

* lg@gzucm.edu.cn

**Data Availability Statement:** All relevant data are within the paper and its Supporting Information files.

## Abstract

Dengue fever is a major worldwide public health problem that, as estimated by the WHO, causes epidemics in over 100 countries, resulting in hundreds of millions of dengue virus (DENV) infections every year. In China, dengue fever mainly occurs in coastal areas. Recurring dengue outbreaks were reported by Guangdong Province almost every year since the first epidemic in 1978. DENV infections persisted in Guangzhou in consecutive years since 2000, with the dengue epidemic reaching a historical peak in 2014. Because Guangzhou is one of the largest cities for opening up in China, understanding the epidemiological characteristics of dengue fever in the city can hopefully provide a significant basis for developing effective dengue prevention strategies. In this study, a total of 34 DENV strains, including 29 DENV-1 strains and 5 DENV-2 strains, were isolated from a blood samples drawn from patients who were diagnosed with dengue fever by hospitals in Guangzhou during 2018. To explore the epidemiological characteristics of dengue fever, the envelope (E) gene obtained from the isolates was amplified for phylogenetic analysis. The results from the phylogenetic analysis showed that DENV in Guangzhou was mainly imported from Southeast Asian countries. Additionally, propagation paths based on phylogeographical analysis suggested potential local dengue transmission in Guangzhou.

## Introduction

Dengue fever, also known as "break-bone fever", is an acute mosquito-borne viral disease caused by dengue virus (DENV) [1]. Patients infected with DENV can have flu-like symptoms, including a high fever, headache, and vomiting, which generally last for ten days. In a small

**Funding:** This work was supported by 81803813, to Li G, from National Nature Science Foundation of China; 201804010029, to Li G, from Science & Technology Planning Project of Guangzhou; SKLNMKF201906, to Li G, from The Open Project of State Key Laboratory of Natural Medicines; A1-AFD018191A17, to Liu XH, from Key Research Projects of GZUCM First-class universities and Top Disciplines; 2018KCXTD007, to Liu XH, from Science & Technology Planning Project of Guangdong Province Office of Education; 2014GKXM032, to Lai XP, from Science & Technology Planning Project of Guangdong Province Office of Education; 81973549, to Li G, from National Nature Science Foundation of China.

**Competing interests:** The authors have declared that no competing interests exist.

proportion of cases, the disease develops into dengue hemorrhagic fever (DHF) or dengue shock syndrome (DSS) [2, 3]. Dengue is principally spread by *Aedes aegypti* and *Aedes albopictus* mosquitoes in tropical and subtropical regions where the warm and humid climate indicates favorable mosquito habitat. In other words, dengue fever is also a seasonal infectious disease, and mosquito densities and climatic conditions are strongly associated with the incidence of dengue fever. In recent decades, dengue fever, as a growing threat to human health, has presented a tremendous challenge to health service providers throughout the world [4]. According to a published study, an estimated 390 million dengue infections occur in over 100 countries every year [3, 5]. Globally, the vast majority of the world population is exposed to DENV, making dengue one of the most dangerous vector-borne viral diseases worldwide [4]. Dengue outbreaks have a devastating effect on public health and economic sectors [3]. In the stricken areas, the disease leads to a slowdown in the local economy as it inflicts a significant health burden on the population, impairs people's quality of life and impedes the development of the tourist industry. With vector management as the primary means to control and prevent DENV transmission, it should be noted that improper use of insecticides is a waste of resources and a source of environmental pollution that threatens both human and environmental health [6, 7]. So it is important to know the structure of dengue virus.

DENV, as a member of the genus *Flavivirus* in the family *Flaviviridae*, is an enveloped, single-stranded, positive-sense RNA virus. The DENV genome is approximately 11,000 nucleotides in length and encodes three structural proteins, namely, the capsid (C), premembrane (prM), and envelope (E) proteins, and seven nonstructural (NS) proteins (NS1, NS2A, NS2B, NS3, NS4A, NS4B and NS5) in a single open reading frame (ORF). Specifically, the 5' end of the DENV genome encodes the C/prM/E proteins, while the 3' end encodes the seven NS protein[8]. There are five DENV serotypes, which are referred to as DENV-1, DENV-2, DENV-3, and DENV-4, and the distinctions between these serotypes are based on their antigenicity. Additionally, different serotypes can be further differentiated into different genotypes[9].

In fact, as early as 1978, a case of imported dengue fever was discovered in Xiamen, after which the Chinese mainland has experienced multiple dengue epidemics in the past few decades [10]. In addition to the coastal provinces in Southeast China and Taiwan that are mainly affected by the disease, an increasing geographic expansion to the inland is noted as the global climate changes [10, 11]. Guangdong Province has been declared a severe epidemic area by the Chinese Center for Disease Control and Prevention, as it has been frequently attacked by dengue fever since the first outbreak [12]. Despite the absence of reported dengue fever cases between 1982 and 1984, 1988 and 1989, and in 1994, 1996, and 1998, the disease has been a constant threat in Guangdong Province [13]. In Guangzhou, the capital city of Guangdong Province, dengue incidence, regardless of the number of DENV infections, has been reported every year since 2000 [14]. In 2014, Guangdong Province experienced its worst dengue outbreak on record, during which different DENV serotypes were detected in the DENV-infected patients [12, 15]. There were over 46000 dengue fever cases notified nationwide throughout the year, including up to 45230 cases and 76 imported cases reported by Guangdong Province, which exceeds the cumulative number of infections between 1990 and 2013 [14, 16].

DENV infections are simply classified as imported and autochthonous cases according to patients' recent travel records. Dengue epidemics refer to continued widespread outbreaks of dengue fever in areas where DENV affects a large population and is transmitted between people by the mosquitoes *Aedes aegypti* and *Aedes albopictus*. To date, there are no other DENV foci in mainland China, except for the reported natural focus in Yunnan Province [17]. As a result of rapid economic growth, thriving tourism, and the greenhouse effect, there were many patients infected with dengue fever in Guangdong Province, and a serious outbreak occurred

in 2014. [18, 19]. The endemicity of dengue in Guangzhou has been heatedly discussed [20]. A previous study based on phylogenetic analysis suggested the presence of local dengue transmission in Guangzhou [10]. In the present study, DENV strains were isolated from blood samples drawn from DENV-infected patients who were diagnosed with dengue fever by hospitals in Guangzhou during 2018. We described the epidemiological characteristics of isolated DENV based on phylogenetic reconstruction and analysis using a dataset comprised of DENV sequences downloaded from the GenBank database and those obtained from genome sequencing.

## Materials and methods

### 1. Sample collection

In this study, all patient samples were approved by the First Affiliated Hospital of Guangzhou University of Chinese Medicine and Guangdong Hospital of Traditional Chinese Medicine. All patients had fever $> 37.5\degree C$ for less than 72 h. Anti-dengue IgM and IgG enzyme-linked immunosorbent assay (ELISA) kits were used to confirm dengue infection [21]. A total of 170 samples suspected of having dengue fever occurred during the 2018 outbreak. 55 blood samples were determined to be positive samples by hospital diagnosis.

Oral informed consent was obtained from all patients who were involved in this study.

### 2. Virus isolation, RNA extraction and serotyping

Dengue virus was isolated from 55 positive blood samples. To increase virus titer, each patient's serum was inoculated into an Aedes albopictus C6/36 cell line cultured in MEM with 10% FBS (Gibco, Carlsbad, CA, USA) and diluted 1:40 in fresh MEM. Next, the medium was added to C6/36 cells and incubated at $30\degree C$ for 2 h. Then, the residual inoculum was replaced with fresh MEM containing 2% FBS; the cells were maintained at $30\degree C$ in a humidified atmosphere of 5% $CO_2$ for 5 to 7 days (until significant cytopathic effects (CPE) occurred). Simultaneously, the supernatants were collected for RNA extraction after centrifugation at 4000 rpm. These samples were stored at $-80\degree C$ until needed.

Viral RNA was extracted from 200 μl of each supernatant using the QIAamp Viral RNA Mini Kit (Qiagen, Germany) as instructed by the manufacturer. Then, the RNA was reverse-transcribed into cDNA by reverse transcriptase at $37\degree C$ for 90 min and then at $70\degree C$ for 10 min. DENV serotyping was carried out by multiplex RT-PCR [22].

### 3. Sequencing

Using RNA as templates, RT-PCR was performed to amplify the envelope (E) gene. DENV-1 primers were designed for sequencing based on the DENV-1 standard Hawaii strain (GenBank accession number: KM204119) and DENV-2 strain (GenBank accession number: KM279569) [10, 22]. E gene sequences of 34 strains were obtained from the patient samples. The primers used in this article are listed in Tables 1 and 2. These sequences were uploaded to the GenBank database, and their IDs, i.e., the GenBank accession numbers of the sequences, are listed in Table 3.

### 4. Dataset

In the dataset, the new isolates and the sequences of DENV-1 and DENV-2 available from the GenBank database were aligned and adjusted using MAFFT v7.308 [23] and Aliview to remove those without a sampling date or geographic location (The length of the E gene sequence is 1485). Recombination Detection Program (RDP v4.36) was employed in recombination

**Table 1. Primer sequence, and size of RT-PCR product and generated combination of primers.**

| Virus | Primer | Primer sequence (from 5′ to 3′) | Size of amplicon &Primer combination | Primer position |
|---|---|---|---|---|
| | Dcon-F | AGTTGTTAGTCTACGTGGACCGACA | | 1–25 |
| DENV-1 | DENV-1-R | CCCCGTAACACTTTGATCGCTCCATT | 342 bp Dcon-F and DEN1-R | 317–342 |
| DENV-2 | DENV-2-R | CGCCACAAGGGCCATGAACAG | 251 bp Dcon-F and DEN2-R | 231–251 |
| DENV-3 | DENV-3-R | GCACATGTTGATTCCAGAGGCTGTC | 538 bp Dcon-F and DEN3-R | 514–538 |
| DENV-4 | DENV-4-R | GTTTCCAATCCCATTCCTGAATGTGGTGT | 754 bp Dcon-F and DEN4-R | 726–754 |

analysis with a wide range of recombination detection methods, including RDP, Chi-maera, BootScan, 3Seq, GENECONV, MaxChi, and SiScan [24]. To avoid redundancy, CD-HIT-EST (http://weizhongli-lab.org/cdhit_suite/cgi-bin/index.cgi?cmd=cd-hit-est) was used to group sequences from outside China into clusters, with the nucleotide identity threshold of 100% [25]. A single sequence was randomly chosen from each cluster as a representative of all sequences that shared the same sampling date and geographic location.

## 5. Phylogenetic signal assessment and nucleotide substitution model selection

The accuracy of phylogenetic inference based on the nucleotide sequence dataset is subject to the saturation level of the phylogenetic signal. Substitution saturation is a crucial factor because a phylogenetic tree becomes meaningless if the aligned sequences in the dataset lose phylogenetic information due to substitution saturation. In the present study, the substitution saturation level of each dataset was assessed with DAMBE and Xia's test method [26].

Models of nucleotide substitution allow for the calculation of probabilities of change between nucleotides along the branches of a phylogenetic tree. On this basis, we used the Akaike information criterion (AIC) to select a best-fit model and created a maximum-likelihood tree (ML tree) with the model. Additionally, a maximum clade credibility (MCC) tree was generated using the Bayesian Information Criterion (BIC) for model selection. The best-fit models of nucleotide substitution were estimated with JModeltest v2.1.7 [27]. Subsequently, the IQ-TREE [28] was employed to determine the best-fit models since a massive quantity of sequences were deposited in a global dataset, which was beyond the capacity of JModeltest.

## 6. Phylogenetic tree reconstruction

In search of more information about the samples, we set up a global dataset that involved all samples and the DENV-1 and DENV-2 sequences available from the GenBank, except those being removed from the database as redundant sequences. Furthermore, ML trees were constructed with best-fit models in RAxML v8.0.9 [29], while the reliability of topology was evaluated using bootstrap values derived from 1000 repetitions. The ML trees were visualized with FigTree v1.4.2.

**Table 2. Primers used for amplifying and sequencing the complete DENV-1and DENV-2 envelope gene.**

| Reaction | Primer name | Sequence (from 5′ to 3′) | Primer position |
|---|---|---|---|
| Amplification & Sequencing | DENV1-E-F | TGCCATAGGAACATCCATCAC | 863–883 |
| | DENV1-E-R | TCCCAATGGCTGCTGATAGTC | 2495–2462 |
| | DENV2-E-F | AATGGCAGCAATCTTGGCATACACC | 747–771 |
| | DENV2-E-R | ACTGAGCGGATTCCACAAATGCCCT | 2504–2480 |

**Table 3. The 2018 Guangzhou dengue virus sequences information isolated and uploaded in this study.**

| ID | Accession | Serotype | Location | Collection date | Cluster |
|---|---|---|---|---|---|
| 1–3 | MK517719 | DENV-1 | Guangzhou | 2018/9/20 | Cluster I |
| 3–8 | MK517720 | DENV-1 | Guangzhou | 2018/9/21 | Cluster I |
| 4–9 | MK517721 | DENV-1 | Guangzhou | 2018/9/21 | Cluster I |
| 5–18 | MK517735 | DENV-1 | Guangzhou | 2018/9/21 | Cluster II |
| 6–19 | MK517740 | DENV-1 | Guangzhou | 2018/9/22 | Cluster I |
| 7–26 | MK517738 | DENV-1 | Guangzhou | 2018/9/22 | Cluster II |
| 8–29 | MK517744 | DENV-1 | Guangzhou | 2018/9/23 | Cluster III |
| 9–32 | MK517739 | DENV-1 | Guangzhou | 2018/9/24 | Cluster II |
| 1–37 | MK517743 | DENV-1 | Guangzhou | 2018/9/24 | Cluster III |
| 2–40 | MK517750 | DENV-2 | Guangzhou | 2018/9/24 | Cluster V |
| 12–43 | MK517746 | DENV-1 | Guangzhou | 2018/9/24 | Cluster III |
| 3–44 | MK517752 | DENV-2 | Guangzhou | 2018/9/25 | Cluster V |
| 12–45 | MK517747 | DENV-1 | Guangzhou | 2018/9/25 | Cluster III |
| 15–48 | MK517734 | DENV-1 | Guangzhou | 2018/9/25 | Cluster II |
| 4–49 | MK517748 | DENV-2 | Guangzhou | 2018/9/25 | Cluster IV |
| 17–59 | MK517722 | DENV-1 | Guangzhou | 2018/9/26 | Cluster I |
| 18–61 | MK517723 | DENV-1 | Guangzhou | 2018/9/26 | Cluster I |
| 19–64 | MK517725 | DENV-1 | Guangzhou | 2018/9/26 | Cluster I |
| 20–66 | MK517727 | DENV-1 | Guangzhou | 2018/9/26 | Cluster I |
| 21–69 | MK517730 | DENV-1 | Guangzhou | 2018/9/26 | Cluster I |
| 22–81 | MK517741 | DENV-1 | Guangzhou | 2018/9/28 | Cluster I |
| 23–84 | MK517729 | DENV-1 | Guangzhou | 2018/9/28 | Cluster I |
| 24–89 | MK517726 | DENV-1 | Guangzhou | 2018/9/28 | Cluster I |
| 25–93 | MK517724 | DENV-1 | Guangzhou | 2018/9/28 | Cluster I |
| 26–94 | MK517731 | DENV-1 | Guangzhou | 2018/9/28 | Cluster I |
| 1–97 | MK517733 | DENV-1 | Guangzhou | 2018/10/1 | Cluster II |
| 27–107 | MK517742 | DENV-1 | Guangzhou | 2018/10/3 | Cluster I |
| 28–108 | MK517745 | DENV-1 | Guangzhou | 2018/10/3 | Cluster III |
| 31–134 | MK517728 | DENV-1 | Guangzhou | 2018/10/5 | Cluster I |
| 32–135 | MK517732 | DENV-1 | Guangzhou | 2018/10/5 | Cluster I |
| 5–141 | MK517751 | DENV-2 | Guangzhou | 2018/10/6 | Cluster V |
| 6–146 | MK517749 | DENV-2 | Guangzhou | 2018/10/6 | Cluster IV |
| 3–169 | MK517736 | DENV-1 | Guangzhou | 2018/10/8 | Cluster II |
| 4–170 | MK517737 | DENV-1 | Guangzhou | 2018/10/8 | Cluster II |

IDs in Table 1 are registration numbers when collecting blood samples; the clustering is distinguished according to ML trees.

## 7. Phylogeographic analyses

According to the ML trees obtained from the global dataset, the newly isolated sample sequences in the DENV-1 and DENV-2 datasets were divided into different clusters (with the classification of clusters based on bootstrap values and the number of 70% as a cut-off for a "reliable" branch). A sample sequence was selected from each cluster for nucleotide BLAST on the NCBI web server, and 100 DENV sequences were downloaded from each BLAST run. On this basis, the sample sequence of Cluster III was set to match 1000 sequences to gather more biological information. The downloaded sequences and the samples were combined together and divided into four datasets for comparison on the MAFFT web server. Phylogenetic signals of these datasets were detected as described above.

Each dataset was analyzed with TempEst v1.5 to investigate its temporal signal and "clockli-keness" of ML phylogenies before applying the assumption of a molecular clock into the phylo-genetic analysis [30]. ML trees generated by RAxML were inputted, and the sample dates of the sequences were defined. After estimation of the best-fitting root, a linear regression was performed on the root-to-tip distances of samples versus the date of the isolate, and significant outliers (sequences) were rejected as they did not fit into the assumption of the molecular clock.

The spatial diffusion of the time-scaled genealogy was modeled as a continuous-time Mar-kov chain (CTMC) process with the program BEAST v1.8.3 [31]. The diffusion process along the phylogenies of the datasets for BLAST was estimated using the Bayesian stochastic search variable selection (BSSVS) procedure. The uncorrelated lognormal relaxed clock (ULRC) method was adopted to produce phylogenetic estimates. In addition, we chose a Bayesian sky-line coalescent model to review the demographic history and ran the Bayesian MCMC chain for a bulk number of iterations to ensure process convergence. The BEAGLE package was used to speed up the calculation process [32]. The effective sample size (ESS) was calculated using TRACER [33], with all parameters showing ESS values > 200 after burn-in in the initial 10% iterations. The MCC tree was summarized using TreeAnnotator v1.8.2 at the burn-in rate of 10% and was visualized with FigTree v1.4. SPREAD v1.0.6 is used to generate the propaga-tion path map. Import the files generated by the BEAST software into the SPREAD software, set the latitude and longitude of the location, and set the BF value to 3, and then generate the corresponding path map.

## 8. Group average distance calculation

We differentiated the DENV-1 and DENV-2 data sets based on genotypes. In order to avoid the impact of old viruses on genetic distance calculation, we only extracted the genotypes of sequence samples from Guangdong Province in the past five years. The MEGA software was used to compare sample sequences of different clusters and compare them to sequences not in the cluster.

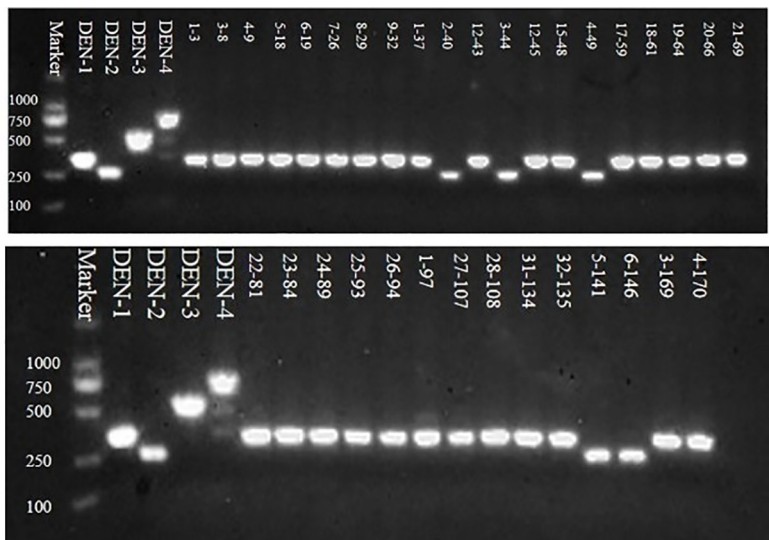

**Fig 1. Gel electrophoresis analysis of RT-PCR products.** According to the results from gel electrophoresis analysis of different samples given in the figure, there are 34 RT-PCR products; the sizes of the amplicons (ID 2–40, 3–44, 4–49, 5–141, and 6–146) are consistent with DENV-2, while the rest are consistent with DENV-1.

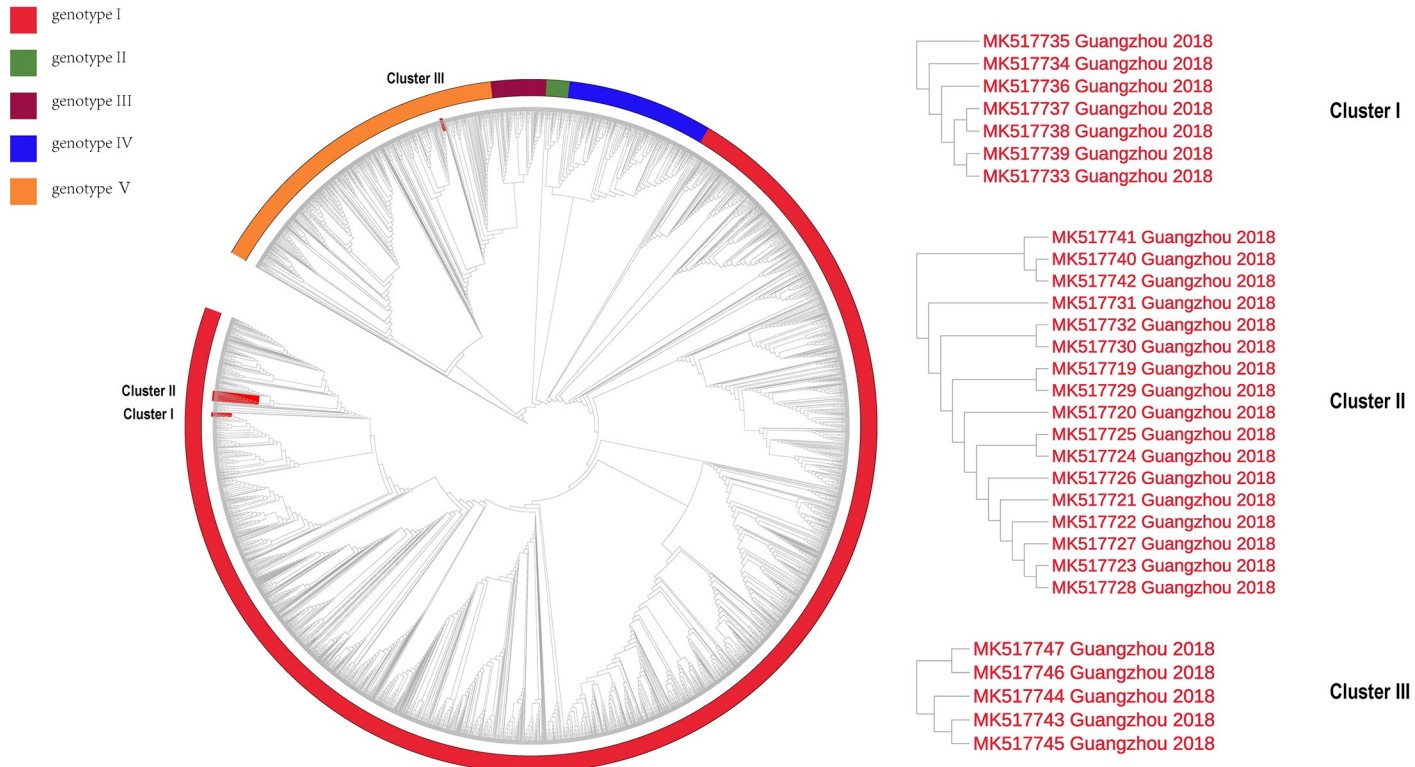

**Fig 2. Maximum likelihood tree for global dataset reconstruction of DENV-1.** The maximum likelihood tree result of DENV-1 is shown in the figure. The sample sequence clusters under different branches, and the red blocks labeled in the branches in the figure are clusters of sample sequence clusters. Different color blocks outside the branches represent different genotypes.

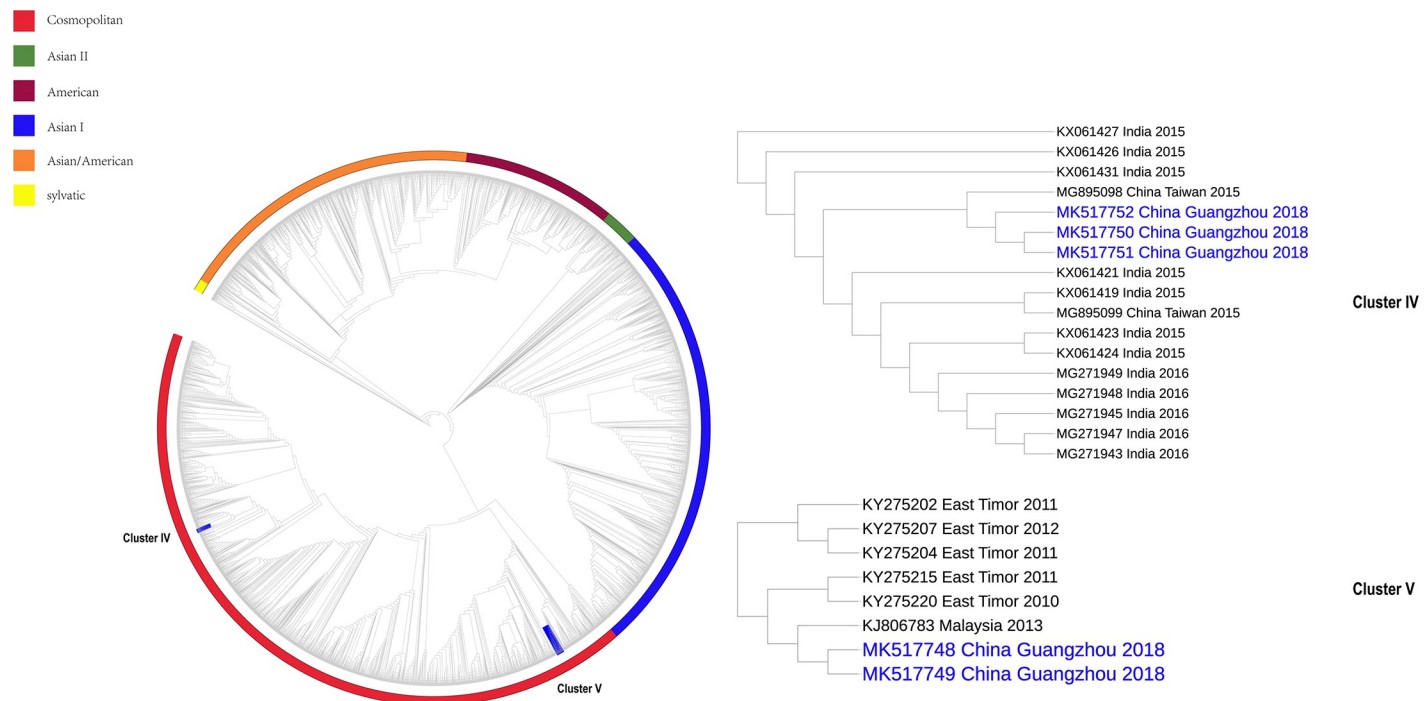

**Fig 3. Maximum likelihood tree for global dataset reconstruction of DENV-2.** The maximum likelihood tree result of DENV-2 is shown in the figure. The sample sequence clusters under different branches, and the blue blocks labeled in the branches in the figure are clusters of sample sequence clusters. Different color blocks outside the branches represent different genotypes.

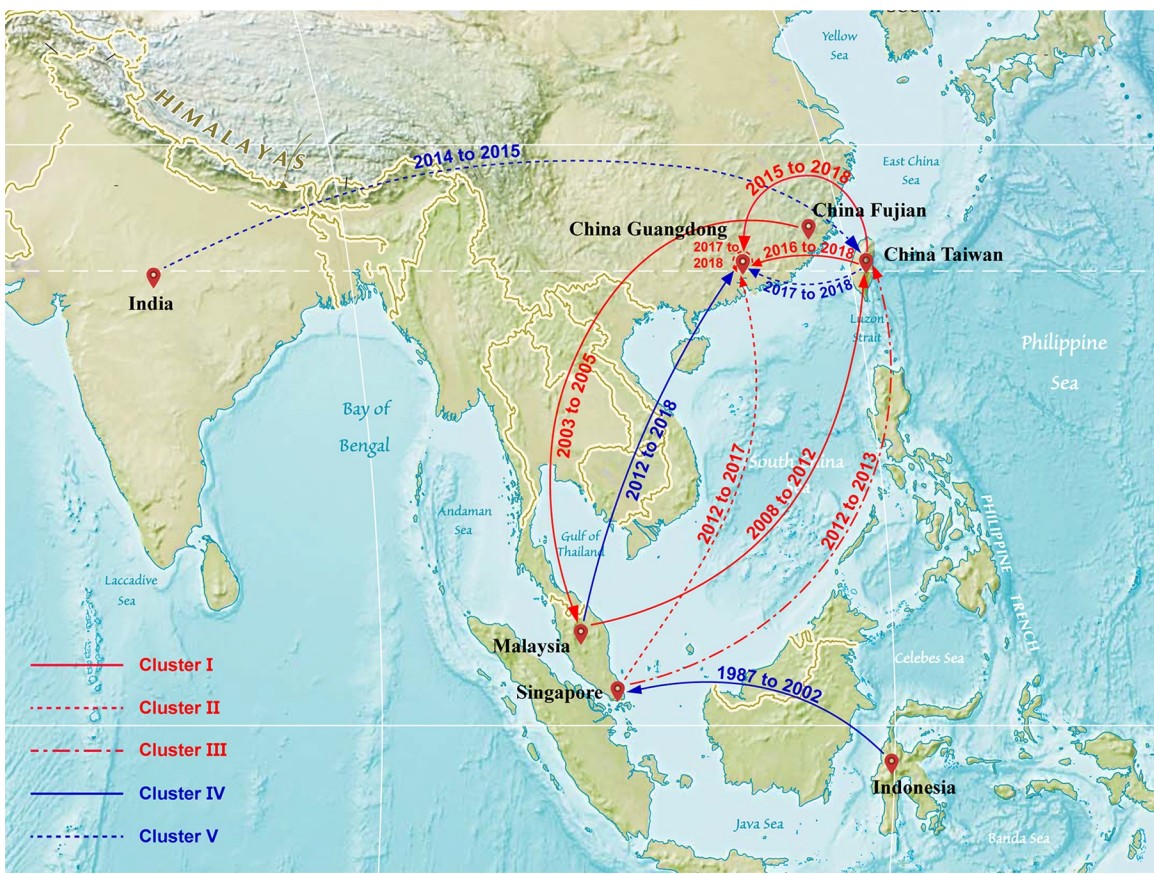

**Fig 4. The propagation path map of dengue virus in Guangzhou in 2018.** The propagation path map is drawn based on the paleogeography analysis. The red curves indicated that the clusters of sample sequences were belonged to DENV-1, while the blue curves belonged to DENV-2, which indicated the propagation path. The time beside the curves indicates the time range in which the virus strain is introduced, and the red flag indicates different locations.

## Result

### 1. Sample information and serotypes

In 2018, a total of 170 blood samples were provided by the First Affiliated Hospital of Guangzhou University of Traditional Chinese Medicine, and 34 strains of dengue viruses were obtained from these blood samples, including 29 strains of DENV-1 and 5 strains of DENV-2. After gel electrophoresis analysis, the RT-PCR products were sequenced and uploaded to the GenBank database to obtain accession numbers. Details are shown in Fig 1 and Table 3. The original image of agarose gel electrophoresis is shown in S1 Fig.

### 2. Phylogenetic signal assessment

The substitution saturation level of each dataset of the DENV E gene was assessed with DAMBE. The results of the assessment indicated that no subset experienced full substitution saturation.

### 3. Phylogenetic analysis and molecular clock tests

Obviously, the sample sequences were classified in independent clusters with high bootstrap values (70–100%), indicating robust support for the tree topology. DENV-1 was used to

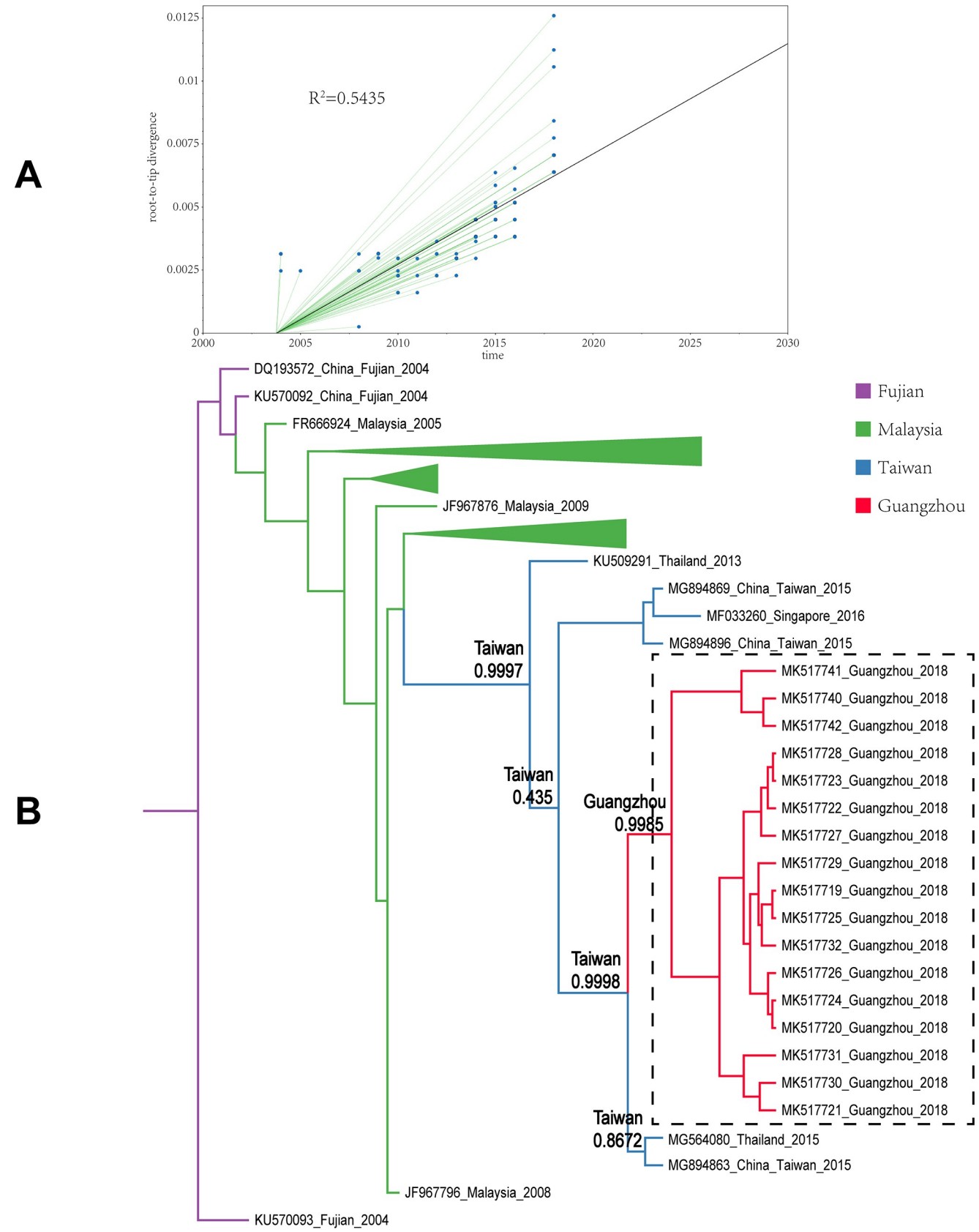

**Fig 5. Maximum clade credibility tree of the sequence of cluster I.** A| A root-to-tip analysis was performed in TempEst v1.5. An ML tree was built using the sample dataset to determine the R2 value; R2 > 0 indicates a positive correlation between the dataset and the molecular clock. B| Sequences from different geographic regions are represented by different colors. The location on the branch is the most recent ancestor position, and the value is the posterior probability. The purple branch indicates that Fujian Province is the most recent ancestor of the sequences; the green branch corresponds to Malaysia; the blue branch corresponds to Taiwan Province; and the red branch corresponds to Guangzhou. The dotted line box represents the sample sequence obtained in this experiment.

construct 3529 sequences of ML tree, and 3542 of DENV-2. From the ML tree of the global dataset, the sample sequences positive for DENV-1 were divided into three clusters, and the DENV-2 positive sample sequences were grouped into two clusters (for detailed clustering results, see Fig 2, Fig 3 and Table 1). Molecular clock tests were performed on the basis of the ML trees. A stable temporal structure was observed as the linear regression on the root-to-tip distances versus the data suggested a relatively high fitting degree, and $R^2$ had a positive value, which enabled further study of the evolutionary process. The molecular clock detection maps are displayed on the corresponding MCC maps.

## 4. Ancestral reconstruction and discrete paleogeography analysis

All sample sequences were clearly clustered into one branch (with the classification of branches based on posterior probabilities and the number of 0.8 and higher as trustworthy). In this study, all sample sequences in the first, second, and third clusters belonged to DENV-1. According to the MCC tree reconstructed from Cluster I, the 17 isolates were clustered in one branch (posterior probability = 0.9985), which is the same as the Thailand and Taiwanese 2015 sequences (posterior probability = 0.9998). Our paleogeography analysis results showed that the sample virus originated in Taiwan. In 2015, the virus affected the region radically and was later imported into Guangzhou between 2015 and 2018. The MCC tree of Cluster II indicated that the sample sequences were clustered in the branch (accession: MG767211) with a sequence obtained from Guangzhou in 2017 (posterior probability = 0.9999), with no other clusters in the same branch. The results from the paleogeography analysis suggested that Guangzhou was the location where the most recent common ancestor was found, while an earlier ancestor appeared in Singapore. The MCC tree of Cluster III suggested that the sample sequences were clustered in the same branch with the Taiwanese 2015 and 2016 sequences, as well as the cluster of the Singaporean 2016 sequences (posterior probability = 0.9992). Through paleogeography analysis, the sample virus might be spread along a propagation path and imported into Guangzhou from Taiwan in 2018 or before. The fourth and fifth clusters belonged to DENV-2. As shown in the MCC tree of Cluster IV, the sample sequences were clustered in the same branch with the Malaysian 2013 sequence (posterior probability = 0.9999), indicating that the most recent ancestor of the sample sequences emerged from Malaysia. According to the propagation path map, the sample virus entered Guangzhou during 2013–2018. The MCC tree of Cluster V suggested that the sample sequences were clustered in the same branch with the Indian 2015 sequences and the Taiwanese 2015 sequence cluster (posterior probability = 0.8465). According to the propagation path, the sample virus was imported into Guangzhou during 2015–2018. It is difficult to infer a highly accurate propagation path as no more sequences are available from the GenBank database. The propagation path map of all sample viruses is included in Fig 4. The MCC trees are shown in Figs 5–9.

## 5. Average genetic distance comparison

The five sample clusters were compared with the genetic distances of the same genotype in clusters in 2014–2018. It can be seen from the table that the sequence of the DENV-2 two-

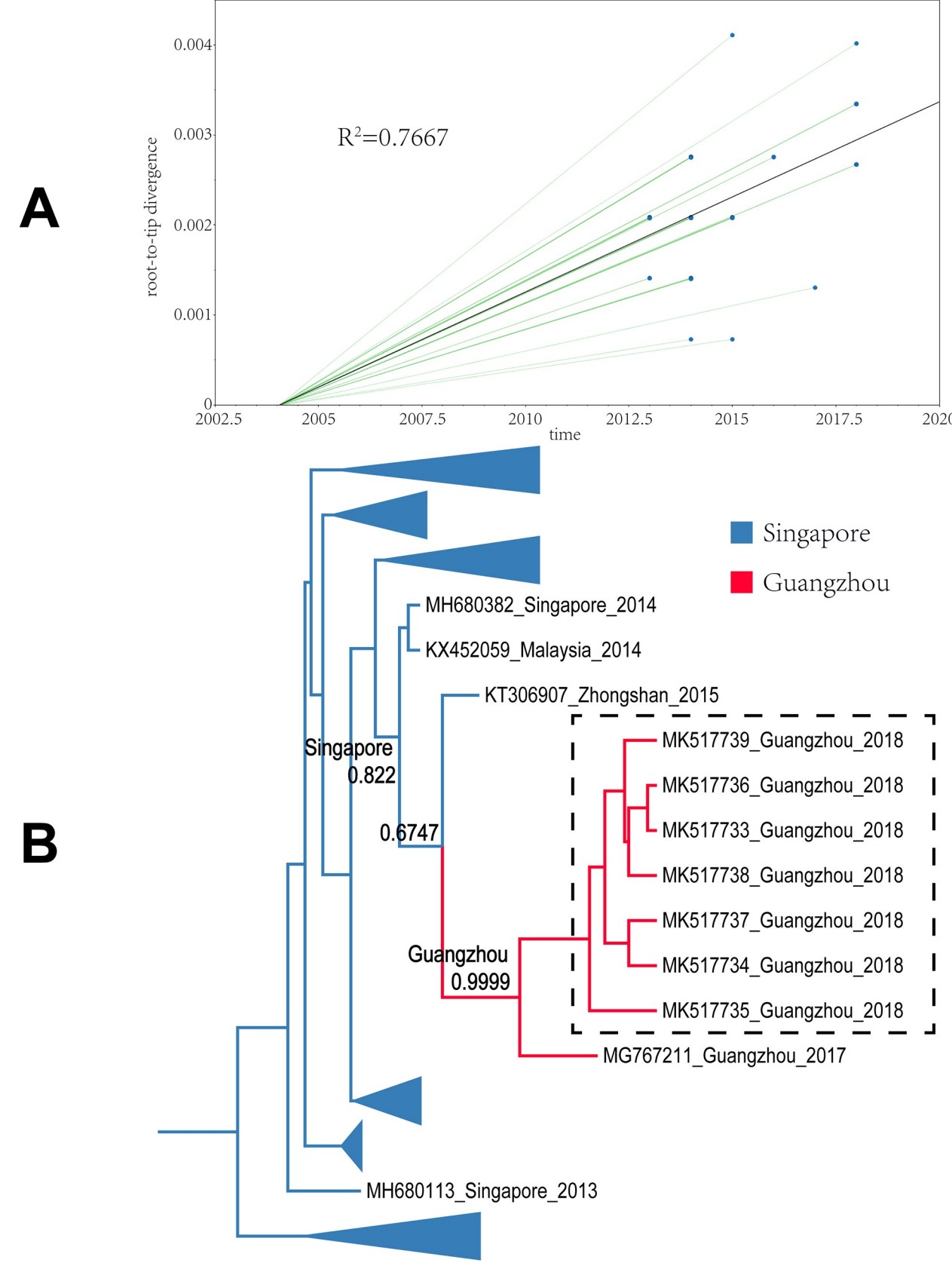

**Fig 6. Maximum clade credibility tree of the sequence of cluster II.** A| A root-to-tip analysis was performed in TempEst v1.5. An ML tree was built using the sample dataset to determine the R2 value; R2 > 0 indicates a positive correlation between the dataset and the molecular clock. B| Sequences from different geographic regions are represented by different colors. The location on the branch is the most recent ancestor position, and the value is the posterior probability. The blue branch indicates that Singapore is the most recent ancestor of the sequences; and the red branch corresponds to Guangzhou. The dotted line box represents the sample sequence obtained in this experiment.

cluster sample collected in this study differs greatly from the sequence of the same genotype. The sequence differences between other sample clusters and the same genotype are small. The data is presented in Table 4.

## Discussion

In the present study, blood samples were drawn from DENV-infected patients who were diagnosed with dengue fever by hospitals in Guangzhou during 2018, and an analysis was performed on the DENV strains isolated from these blood samples. Through gel electrophoresis analysis, the isolated sequences were classified into two serotypes, namely, DENV-1 and DENV-2. Furthermore, DENV-1 strains were divided into three clusters, and the DENV-2 strains were divided into two clusters according to the ML trees. The ancestral reconstruction analysis showed that the sample sequences largely originated from Indonesia, Malaysia, Singapore, and Taiwan. This suggests that the dengue virus that we are popular in Guangzhou is still based on input. As the capital city of Guangdong Province and the trade center in Southeast China, Guangzhou is exposed to a relatively high risk of the disease, as DENV may enter the city along with imported goods and migrant workers. In addition, DENV continuously flows into China because Thailand, Malaysia, Singapore, and other Southeast Asian countries are popular destinations for Chinese tourists, especially those living in Guangdong Province. Therefore, entry-exit inspection and quarantine should be implemented effectively for dengue prevention and control. For instance, suspected and confirmed cases of DENV infection should be isolated and treated properly to reduce the risk of imported dengue fever. In addition, health education also plays an important role in dengue prevention and control. To reduce the risk of DENV infection, public health authorities should provide the necessary materials for tourists to gain a better understanding of dengue prevention and remind them not to visit an endemic area during epidemic seasons.

Interestingly, the results from the traceability analysis showed that the most recent common ancestor of the sequences of cluster II emerged from Guangzhou, and Cluster II and the Guangzhou 2017 isolate are in the same branch without any other foreign sequences. Moreover, they shared a common ancestral lineage to the Malaysian 2014 isolate, the Singaporean 2014 isolate, and the Zhongshan 2015 isolate in other branches. Phylogeographically, there are two possible propagation paths. First, the virus originated in Singapore and became an epidemic in the country before 2014; during the next year, it was imported into Zhongshan, and then in 2017, the disease flowed into Guangzhou from other cities in Guangdong Province; the Guangzhou 2018 isolates came from the virus imported into the city during 2017 as a result of local transmission. Second, the virus emerged from Singapore and was not introduced into Zhongshan until 2015; during 2017 and 2018, the virus was continuously imported into Guangzhou from Singapore or other countries and regions; the sample sequences were not clustered into a branch with the isolates from other countries because no related sequences were available in GenBank or the database had no sufficient patient isolates. Regarding the sources of spread or propagation, some DENV strains were imported from endemic countries and regions and caused dengue fever without further propagation in China; some evolved from imported strains and led to local epidemics during the year; others were localized strains after vertical transmission. Although no substantial evidence was found in this study to prove

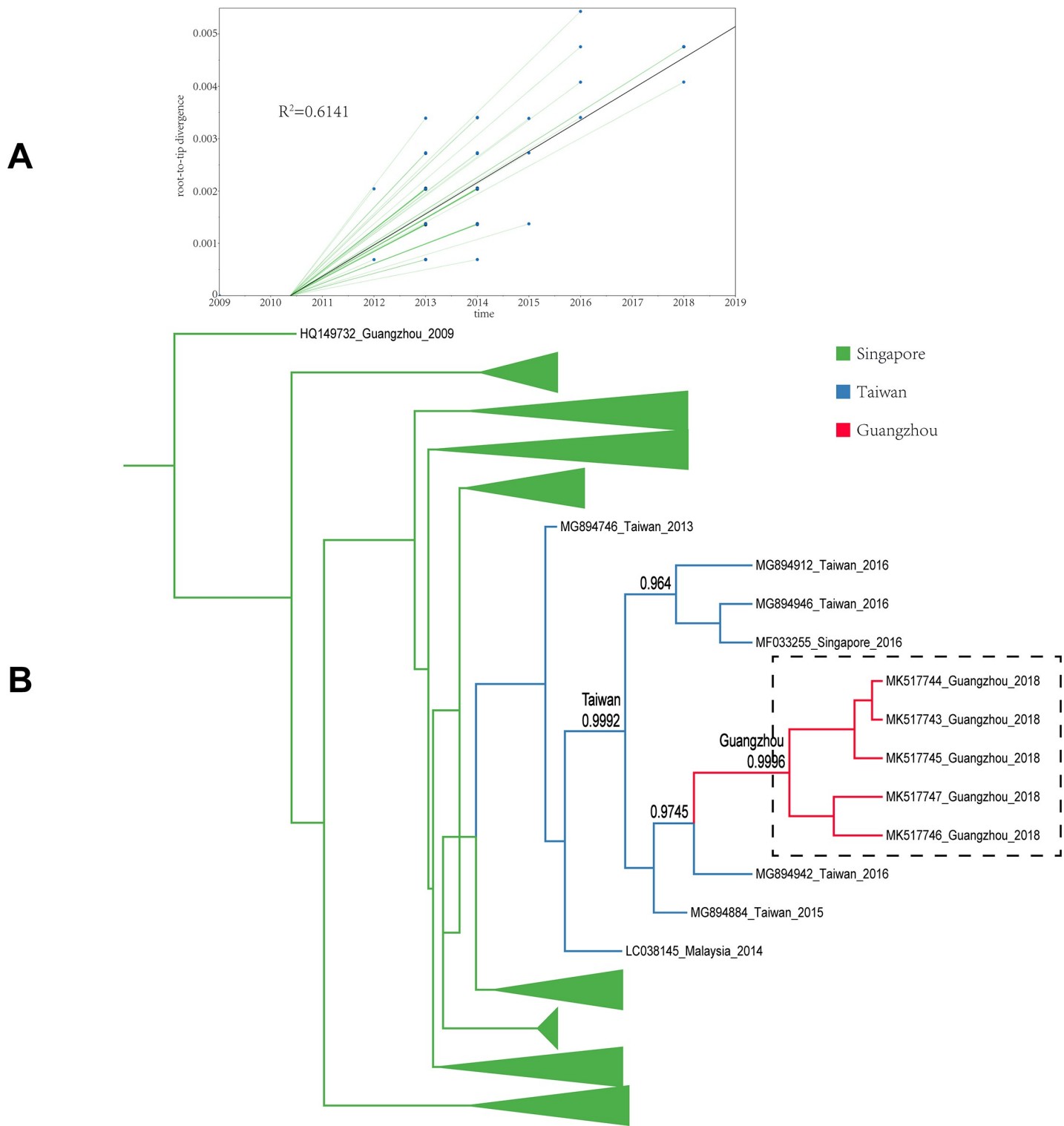

**Fig 7. Maximum clade credibility tree of the sequence of cluster III.** A| A root-to-tip analysis was performed in TempEst v1.5. An ML tree was built using the sample dataset to determine the R2 value; R2 > 0 indicates a positive correlation between the dataset and the molecular clock. B| Sequences from different geographic regions are represented by different colors. The location on the branch is the most recent ancestor position, and the value is the posterior probability. The green branch indicates that Singapore is the most recent ancestor of the sequences; the blue branch corresponds to Taiwan Province; and the red branch corresponds to Guangzhou. The dotted line box represents the sample sequence obtained in this experiment.

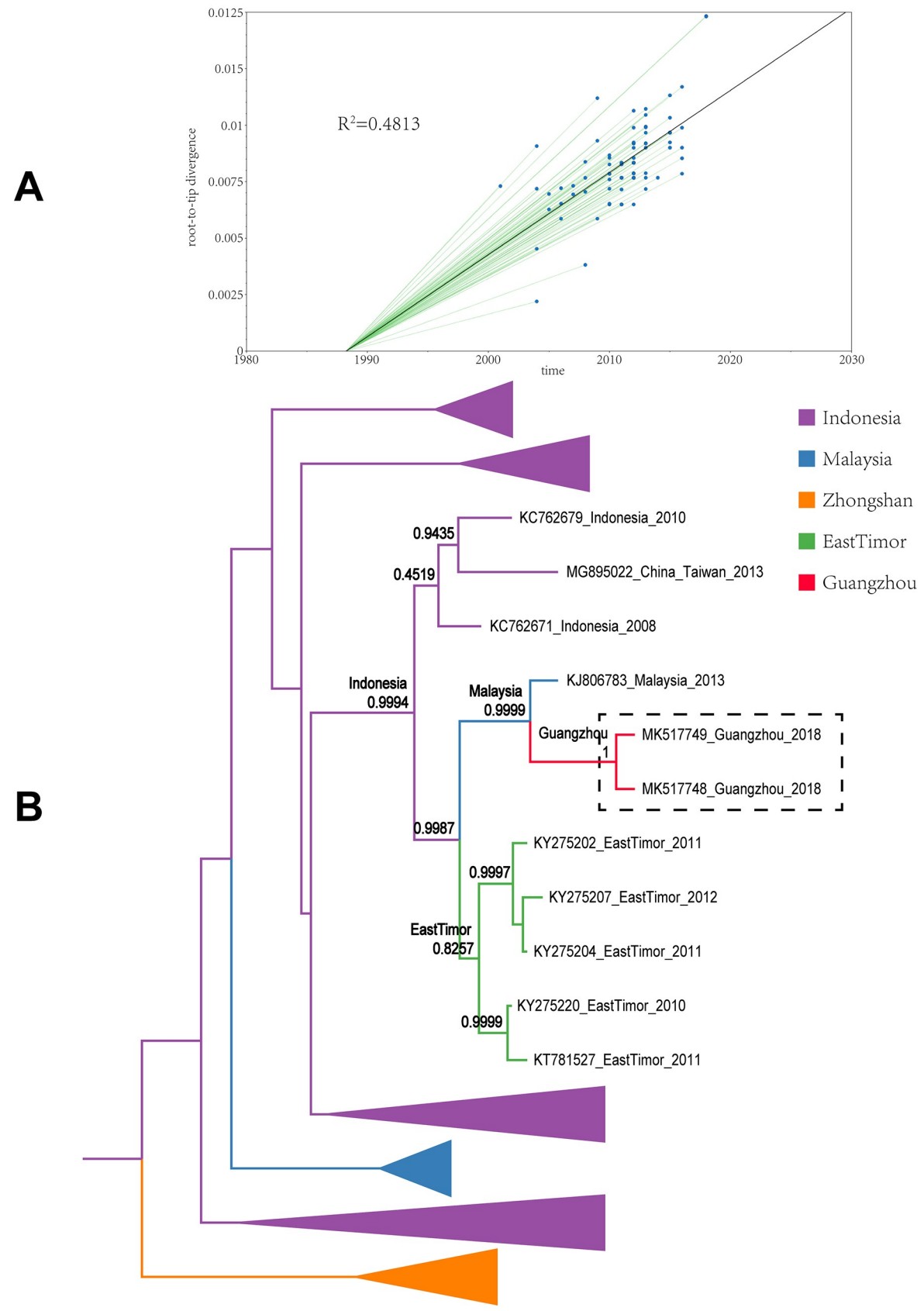

**Fig 8. Maximum clade credibility tree of the sequence of cluster IV.** A| A root-to-tip analysis was performed in TempEst v1.5. An ML tree was built using the sample dataset to determine the R2 value; R2 > 0 indicates a positive correlation between the dataset and the molecular clock. B| Sequences from different geographic regions are represented by different colors. The location on the branch is the most recent ancestor position, and the value is the posterior probability. The purple branch indicates that Indonesia is the most recent ancestor of the sequences; the blue branch corresponds to Malaysia; the orange branch corresponds to Zhongshan City; the green branch corresponds to East Timor; and the red branch corresponds to Guangzhou. The dotted line box represents the sample sequence obtained in this experiment.

vertical transmission of DENV in Guangzhou, previous phylogenetic analyses indicated possible local transmission of dengue fever in the city.

Since *Aedes albopictus* is the main medium for DENV transmission, the localization of DENV largely depends on the formation of localized and virus-carrying eggs and the survival of DENV in the eggs and offspring mosquitoes through winter[34]. A previous study demonstrated that DENV can survive in eggs and spread to offspring mosquitoes. Transovarial transmission of DENV is strongly temperature-related as it determines whether the eggs and young offspring can wait out the winter season. As an example, Yunnan Province has reported the localization of DENV as it has a subtropical monsoon climate and provides a natural habitat for *Aedes albopictus*. Similar to Yunnan Province, Guangzhou has hot, humid summers and mild, dry winters, which create a favorable environment for the breeding of *Aedes* mosquitoes. Moreover, as the greenhouse effect continues to warm the planet slowly, overwintering becomes easier for mosquitoes. Peri-urban areas in Guangzhou have plenty of dirty gullies, open spaces, and rented houses that require effective management to improve the living environment. Additionally, local residents who like indoor and outdoor planting, fish farming, and water harvesting also provide places for mosquito breeding. To reduce mosquito breeding sites, effective urban sanitation management should be implemented. For example, stagnant water should be cleared in a timely manner to ensure the smooth operation and maintenance of the city's drainage system; in crowded places, mosquito prevention and control measures should be taken for the public good. Personal hygiene also plays a critical role in dengue prevention. Residents should avoid keeping fish and aquatic plants and regularly clean up stagnant water in their houses. An individual with a fever or other dengue fever-like symptoms should promptly seek medical attention. Hospitals should strive to improve the diagnostic accuracy and efficiency of dengue fever. Suspected DENV-infected patients should be isolated to prevent further transmission. We should pay closer attention to localized DENV in Guangzhou and strictly implement relevant dengue prevention and control measures.

Although the results of this study implied that vertical transmission of DENV might exist in Guangzhou, there is no solid evidence supporting the inference as no adequate sequences are available for analysis. The coverage of the sequence has a great impact on the study. Patients with dengue sometimes have negative infections. The patient is not aware that the infection with dengue virus has led to a decrease in the reported sequence, and a small number of viruses have not been sequenced and uploaded to the database. These can all lead to the wrong propagation path. Therefore, local centers for disease control and prevention should work closely with disease control and prevention departments of Southeast Asian countries to observe how DENV is prevalent in these countries, thereby identifying sources of propagation for DENV in China and exploring the epidemiological characteristics of dengue fever to provide a basis for dengue prevention strategies. As the Belt and Road Initiative moves forward, Asian, European and African countries expect to increase transport connections for international trade, strengthen people-to-people exchanges and remove barriers to investment and trade. Given that, the conventional entry-exit inspection and quarantine divided by national administrative regions is falling short, leading to an increased risk of dengue and Zika fever in the countries

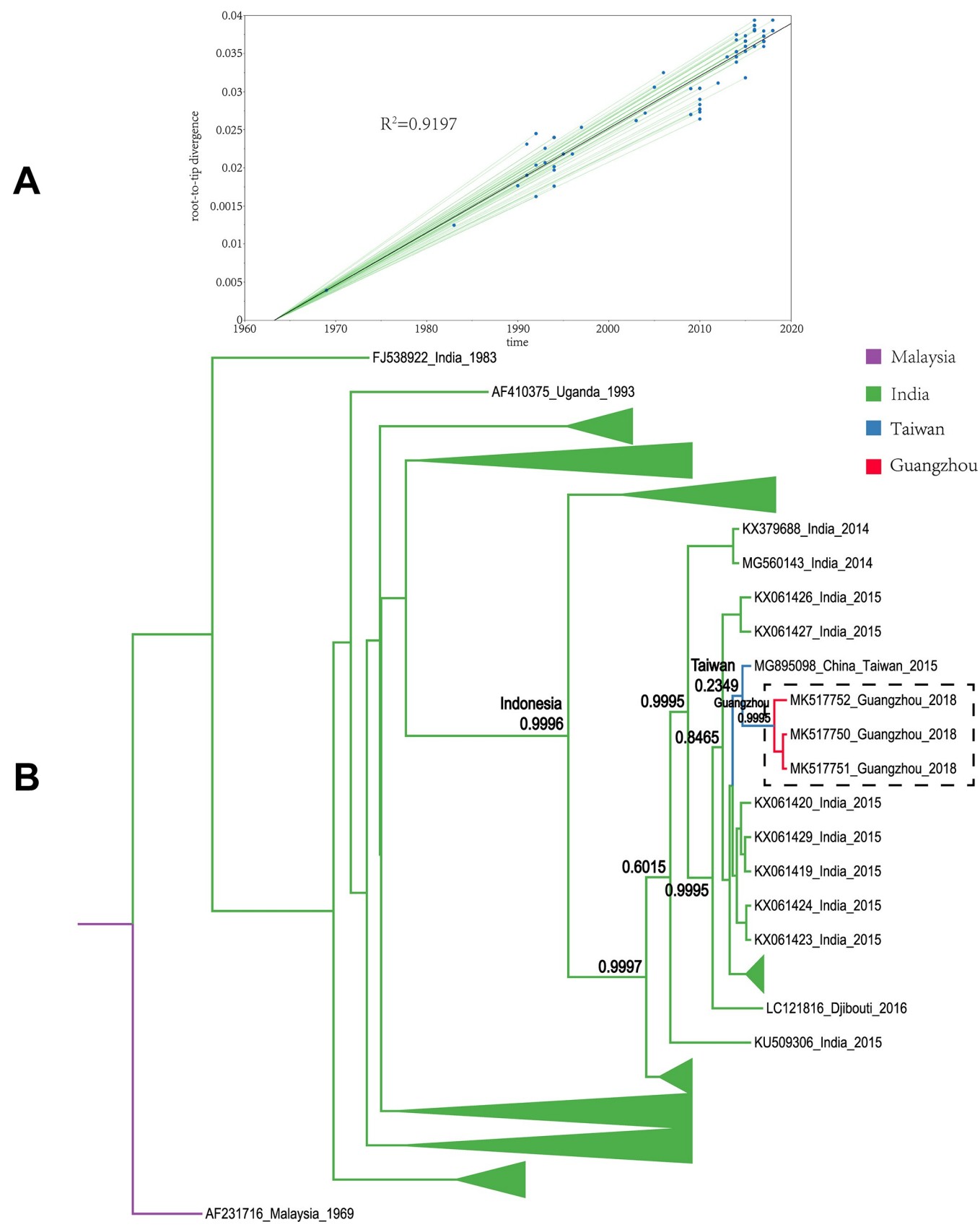

**Fig 9. Maximum clade credibility tree of the sequence of cluster V.** A| A root-to-tip analysis was performed in TempEst v1.5. An ML tree was built using the sample dataset to determine the R2 value; R2 > 0 indicates a positive correlation between the dataset and the molecular clock. B| Sequences from different geographic regions are represented by different colors. The location on the branch is the most recent ancestor position, and the value is the posterior probability. The purple branch indicates that Malaysia is the most recent ancestor of the sequences; the green branch corresponds to India; the blue branch corresponds to Taiwan province; and the red branch corresponds to Guangzhou. The dotted line box represents the sample sequence obtained in this experiment.

**Table 4. Comparison of average genetic distances of different clusters.**

|  | Species 1 | Species 2 | Distance |
|---|---|---|---|
| DENV 1-Genotype I | others-1 | sample2 | 0.0205329387 |
|  | others-1 | sample1 | 0.0189086126 |
|  | sample2 | sample1 | 0.0202604556 |
| DENV 1-Genotype V | others-2 | sample3 | 0.0067526479 |
| DENV 2-Cosmopolitan | others-3 | sample4 | 0.0336621574 |
|  | sample4 | sample5 | 0.0670677460 |
|  | sample5 | others-3 | 0.0657505057 |

Others in the table indicates the non-sample sequence of the genotype in Guangdong Province from 2014 to 2018.

along the Belt and Road. There is an urgent need to raise public awareness about disease prevention and control, as well as intensified measures against dengue and Zika fever. It is necessary to establish joint prevention and control mechanisms for infectious diseases so that China and other relevant countries can work together to effectively combat infectious diseases and safeguard health security with their technologies and resources.

## Supporting information

**S1 Fig. Original image of agarose gel electrophoresis.**
(TIF)

**S1 Table. The kit for diagnosing dengue patients is used in hospitals.**
(XLSX)

**S2 Table. All accession numbers of the DENV-1 sequence used in this study.**
(XLSX)

**S3 Table. All accession numbers of the DENV-2 sequence used in this study.**
(XLSX)

## Acknowledgments

The First Affiliated Hospital of Guangzhou University of Chinese Medicine needs to be thanked for their support. The authors also need to be thanked for the contribution of this study.

## Author Contributions

**Data curation:** Feng Liao.

**Formal analysis:** Feng Liao.

**Investigation:** Feng Liao, Shaofeng Zhan, Pan Pan, Zizhao Lao, Lupin Lin, Yanni Lai, Jianguo Wu.

**Methodology:** Feng Liao, Huini Chen, Jieliang Xie, Yaohua Fan, Shuangfeng Lin, Xiaohong Liu.

**Writing – original draft:** Feng Liao.

**Writing – review & editing:** Feng Liao, Geng Li.

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
