## [Decision Letter · Decision Letter 0]

7 Aug 2019

PONE-D-19-19136

Molecular epidemiological characteristics of dengue virus carried by 34 patients in Guangzhou in 2018

PLOS ONE

Dear Dr. Li,

Thank you for submitting your manuscript to PLOS ONE. After careful consideration, we feel that it has merit but does not fully meet PLOS ONE’s publication criteria as it currently stands. Therefore, we invite you to submit a revised version of the manuscript that addresses the points raised during the review process.

We would appreciate receiving your revised manuscript by Sep 21 2019 11:59PM. To enhance the reproducibility of your results, we recommend that if applicable you deposit your laboratory protocols in protocols.io, where a protocol can be assigned its own identifier (DOI) such that it can be cited independently in the future. For instructions see: http://journals.plos.org/plosone/s/submission-guidelines#loc-laboratory-protocols

We look forward to receiving your revised manuscript.

Kind regards,

Jason Blackard, PhD

Academic Editor

PLOS ONE

Journal Requirements:

2. Thank you for stating the following in the Acknowledgements Section of your manuscript:

[The research work was supported by the National Nature Science Foundation of China (81803813),

376 the Science & Technology Planning Project of Guangzhou (No.201804010029), the Special Foundation

377 supported by the Central Government for Higher Education Research Program (2013-2015): Project

378 Number: A1-AFD016141A02-14, A1-AFD018181A28) and the State Major Research Program for

379 Innovation Team of Guangdong Province (Project Number: E1-KFD015151K02).]

We note that you have provided funding information that is not currently declared in your Funding Statement. However, funding information should not appear in the Acknowledgements section or other areas of your manuscript. We will only publish funding information present in the Funding Statement section of the online submission form.

 [The funders had no role in study design, data collection and analysis, decision to publish, or preparation of the manuscript.]

4. We note that Figure [3] in your submission contains a map image which may be copyrighted. All PLOS content is published under the Creative Commons Attribution License (CC BY 4.0), which means that the manuscript, images, and Supporting Information files will be freely available online, and any third party is permitted to access, download, copy, distribute, and use these materials in any way, even commercially, with proper attribution. For these reasons, we cannot publish previously copyrighted maps or satellite images created using proprietary data, such as Google software (Google Maps, Street View, and Earth). For more information, see our copyright guidelines: http://journals.plos.org/plosone/s/licenses-and-copyright.

You may seek permission from the original copyright holder of Figure(s) [3] to publish the content specifically under the CC BY 4.0 license. 

If you are unable to obtain permission from the original copyright holder to publish these figures under the CC BY 4.0 license or if the copyright holder’s requirements are incompatible with the CC BY 4.0 license, please either i) remove the figure or ii) supply a replacement figure that complies with the CC BY 4.0 license. Please check copyright information on all replacement figures and update the figure caption with source information. If applicable, please specify in the figure caption text when a figure is similar but not identical to the original image and is therefore for illustrative purposes only.

Additional Editor Comments (if provided):

This a phylogenetic study of dengue virus in China.  The population size is quite small; therefore, robust inferences are very difficult.  The phylogenetic analyses are well described and appropriate for this type of study.

It would have been very helpful for the authors to evaluate dengue virus from other years as well.  Restricting the study to a single year – even an outbreak year – limits usefulness of the data collected.  For instance, did the virus evolve in 2018 in such a way as to be different from previous years?  If so, what are the virologic differences across the viral genome?

Are other dengue virus sequences available from southern China in the years before 2018?  These should be included in the phylogenetic analyses.

Serum samples from 170 individuals with dengue were evaluated, so why are genotypic data presented for only 34?  How are these 34 different / similar to the 55 individuals with positive blood samples?

How large are the E gene sequences that were analyzed?  This should be stated explicitly.

Lines 150-152 state that a set of global dengue references were used.  How many DENV-1 and DENV-2 references were included?

How was the propagation path evaluated?  This part of the analysis is not described in the Methods.

For the 5 sequence clusters, the authors should calculate the median genetic distance and compare that to sequences that are not in clusters.  It is important for the readers to have some quantified measure of similarity / dissimilarity amongst sequences within a cluster.

Reviewers' comments:

Reviewer's Responses to Questions

**Comments to the Author**

1. Is the manuscript technically sound, and do the data support the conclusions?

Reviewer #1: No

Reviewer #2: Partly

2. Has the statistical analysis been performed appropriately and rigorously? 

Reviewer #1: Yes

Reviewer #2: No

3. Have the authors made all data underlying the findings in their manuscript fully available?

Reviewer #1: Yes

Reviewer #2: No

4. Is the manuscript presented in an intelligible fashion and written in standard English?

Reviewer #1: Yes

Reviewer #2: Yes

5. Review Comments to the Author

Reviewer #1: Liao et al. studied 34 DENV strains, including 29 DENV-1 strains and 5 DENV-2 strains, isolated from blood samples from dengue fever patients at Guangzhou in 2018. They amplified the envelope genes of these isolates by RT-PCR and performed phylogenetic analysis. The found DENV isolates of 2018 in Guangzhou were mainly imported from Southeast Asian countries. Propagation paths based on phylogeographical analysis suggested potential local DENV transmission in Guangzhou.

The authors have carried out RT-PCR, sequencing and phylogenetic analysis of E genes of 34 DENV strains isolated from DF patients at Guangzhou in 2018. Overall, the observation that DENV isolates in Guangzhou were mainly imported from Southeast Asian countries did not bring new insights to the field. The evidence of possible local DENV transmission in Guangzhou over the years was weak. Moreover, there several places in the introduction, methods and results need to be clarified. Several references were inappropriate or irrelevant. See specific comments. There should be addressed to improve this manuscript.

Specific comments:

1) Line 28: “, including 29 DENV-1 strains and 5 DENV-2 strains, isolated from a blood sample from…”? Should be blood samples.

2) Lines 55-59: “DENV, as a member of the genus Flavivirus in the family Flaviviridae, is an enveloped, single stranded, positive-sense RNA virus [8]. The DENV genome is approximately 11,000 nucleotides in length and encodes three structural proteins, namely, the capsid (C), premembrane (prM), and envelope (E) proteins, and seven nonstructural (NS) proteins (NS1, NS2A, NS2B, NS3, NS4A, NS4B and NS5) in a single open reading frame (ORF) [9,10].” References 8, 9 and 10 are not appropriate for the citation here.

3) Lines 60-63: “There are five DENV serotypes…classified into different subtypes [11]”? Five DENV serotypes are controversial due to lack of evidence in humans. Major literature states 4 DENV serotypes. There is no citation of 5 serotypes and Reference 11 is inappropriate here. Different subtypes?

4) Line 83: “Guangdong Province sees a continuous increase in the number of patients”?

5) Lines 69 to 77: They provided a review of DENV epidemiology in Guangdong up to 2014. What happened between 2015 and 2017 before their study in 2018?

6) Lines 99-100: “IgM and IgG enzyme-linked immunosorbent assay (ELISA) kits were used to confirm dengue infection”. Please describe which commercial kits were used or in-house assays.

7) Lines 100-101: “A total of 170 serum samples were available”. These are likely acute samples. Which days post onset of symptoms were the samples collected?

8) Lines 112-113: “The amplified PCR products…” What are the PCR conditions? Reference 24 was real-time RT-PCR and not relevant here.

9) Line 117: “DENV-2 standard Hawaii strains”?

10) Table 1 “Primer sequence, and size of RT-PCR product…” There is no information of the size of RT-PCR product. At least, the genome positions of primers should be presented. There is no reference of these.

11) Table 2: The genome positions of primers should be presented.

12) Lines 150-152: “…we set up a global dataset that involved all samples and DENV-1 and DENV-2 sequences available from the GenBank…”? How many sequences from the GenBank were selected in the analysis and what are selection criteria? Do they include representative sequences from different genotypes within each serotype?

13) Lines 201-202: “… the entropy-based index presented by Xia. The ….Iss<iss.c)..” by="" entropy-based="" index="" is="" presented="" what="" xia="">14) Lines 336-338: “Since Aedes albopictus is the main medium for DENV transmission…eggs and offspring through winter [34]”. Aedes albopictus is the main medium for DENV? Reference 34 is about program interface and is an irrelevant citation.

15) Lines 332-333: “Others were localized strains after vertical transmission”. Vertical transmission has been reported to be very insufficient. Is vertical transmission the only mechanism to explain localized strains? How about silent transmission or sporadic cases plus under-reporting?

Reviewer #2: Liao et al . determined 34 DENV (29 DENV-1 and 5 DENV-2) sequences of E region from isolated virus in Guangzhou, Guangdong Province, China during 2018. They concluded that DENV in Guangzhou was mainly imported from Southeast Asian countries from the phylogenetic analysis but there are several points to be clarified before they reach to the conclusion.

Major concerns.

1. DENV-1 is sub-divided into five genotypes: U, II II, IV and V. DENV-2 is classified as six genotypes: Asian I, Asian II, Asian/American, Cosmopolitan, American and sylvatic. These genotypes should be assigned in Figure 2 and specify the genotypes the presented cluster I-V belongs to.

2. In Table 3, do location columns really indicate the place of infection?

Is there any questioner asking travel history?

3. There are several reports from China. (1) Lin F et al. (https://doi.org/10.1007/s00705-019-04266-1) already reported the DENV-2 genotype Cosmopolitan in same Guangdong province in 2015. (2) Yu H et al. https://doi.org/10.1038/s41598-019-43560-5.) also reported DENV-2 genotype Cosmopolitan in Zhejiang Province, in 2017. (3) Cao et al. (https://doi.org/10.1371/journal.pone.0213353) reported DENV-1 genotype I and V in 2015. (4) Zou et al. (https://doi.org/10.1371/journal.pntd.0007202) also reported DENV-1 genotype I and V in 2014. Many Asian I viruses were deposited to NCBI from China. Before reached to the conclusion that the origin of the virus isolated in this study derived from Southeast Asian countries, the author should pay more attention to the pre-existing Chinese viruses in phylogenetic analysis. Can you deny the possibility from Yunnan or other part of China?

4. The sequences used in molecular clock tests were deviated to recent sequences.

5. Did the author calculate the Bayes factor for phylogeographic reconstruction in Figure 3? What was the cut-off value?

6. In line 173, why the ULRC was selected?

7. In line 174, why the Bayesian skyline coalescent model was selected?

Minor points:

1. The lines in the phylogenetic tree were faint in Figure 2.

2. In line 61, the reference is required for DENV-5.

3. The accession numbers of the reference sequences used in Figure 2-8 should be listed in the supplemental tables.

 </iss.c)..”>

6. PLOS authors have the option to publish the peer review history of their article (what does this mean?). If published, this will include your full peer review and any attached files.

Reviewer #1: No

Reviewer #2: No

---

## [Author Response · Author response to Decision Letter 0]

6 Oct 2019

Additional Editor Comments

1. It would have been very helpful for the authors to evaluate dengue virus from other years as well. Restricting the study to a single year – even an outbreak year – limits usefulness of the data collected. For instance, did the virus evolve in 2018 in such a way as to be different from previous years? If so, what are the virologic differences across the viral genome?

Answer: We calculated the genetic distance between the sample cluster and the non-sample cluster sequences of Guangdong Province in 2014-2018 and defined them to be compared under the same genotype. The results show that their genetic distance differences are not very significant. The dengue virus has been imported into Guangzhou for a long time, and the genetic distance difference between the sample clusters is not obvious. The dengue virus that is popular in Guangzhou is still based on input. 

2. Are other dengue virus sequences available from southern China in the years before 2018?

These should be included in the phylogenetic analyses

Answer: Yes, we take your questions into account when building the ML tree. After searching for the sequence of dengue virus in Genbank, we separated the sequence into two data sets, DENV-1 and DENV-2. Sequences in the dataset are sorted out to remove invalid sequences (E gene is incomplete, no separation sites, no separation time, etc.) so that our dataset can contain many sequences from different countries. Our dataset contains sequences from other southern China, such as Yunnan (another region in China where dengue epidemics), Fujian, Taiwan, etc. The results of the analysis show that there are no clusters in the second cluster sequence that have other sequences in other provinces in China, and the calculation of the ancestral position also shows this result. The ancestors of other sample clusters showed that most of them were imported from Southeast Asian countries, which also indicates that the dengue virus prevalent in Guangzhou is still dominated by input. 

3. Serum samples from 170 individuals with dengue were evaluated, so why are genotypic data

presented for only 34? How are these 34 different / similar to the 55 individuals with positive

blood samples?

Answer: This is not clearly described in the manuscript.. We collected 170 samples of suspected dengue patients from the hospital (not yet diagnosed), and then the hospital tested the 170 samples and determined that 55 of them were positive samples. We cultured 55 samples according to the hospital's test, and finally obtained only 34 dengue viruses and conducted subsequent studies. The changed part is in the 101-103 lines of the manuscript.

4. How large are the E gene sequences that were analyzed? This should be stated explicitly.

Answer: The E gene sequence was used for the study length of 1485. We have already added in the manuscript. The changed part is in the 130 lines of the manuscript.

5. Lines 150-152 state that a set of global dengue references were used. How many DENV-1

and DENV-2 references were included?

Answer: As with the answer to the second question, we downloaded a large number of sequences and processed them. The data set used to construct the ML tree of DENV-1 contains a total of 3529 sequences. The data set used to construct the ML tree of DENV-2 contains a total of 3542 sequences. The sequences number used to construct the ML tree will be written in the supplemental material. Already added this data in the article. The changed part is in the 214-215 lines of the manuscript.

6. How was the propagation path evaluated? This part of the analysis is not described in the

Methods.

Answer: The propagation path map is the result of the phylogeographic analyses. The BEAST software is used to calculate the nearest common ancestor position of each sequence, and the result is a "tree" file, a "log" file, and the like. SPREAD v1.0.6 software can read these files and generate a propagation path map. Related steps have been added to the manuscript. The changed part is in the 182-184 lines of the manuscript. 

7. For the 5 sequence clusters, the authors should calculate the median genetic distance and

compare that to sequences that are not in clusters. It is important for the readers to have

some quantified measure of similarity / dissimilarity amongst sequences within a cluster.

Answer: This is a very good suggestion. In order to let readers know more about the dengue virus that was popular in Guangzhou in 2018, we used MEGA to analyze their genetic distance. First, we differentiated the DENV-1 and DENV-2 data sets based on genotypes. In order to avoid the influence of the old virus on the calculation of genetic distance, we only extracted the sequence of Guangdong Province for the past five years for the genotype of the sample sequence. Experimental methods and results are provided in the manuscript. The changed part is in the 185-189 lines and 316-319 lines of the manuscript.

Reviewer #1

1. Line 28: “, including 29 DENV-1 strains and 5 DENV-2 strains, isolated from a blood sample from…”? Should be blood samples.

Answer: This is a written error that has been corrected in the manuscript. “blood sample” has been modified to “blood samples” in the assay. We collected 170 samples of suspected dengue patients from the hospital (not yet diagnosed), and then the hospital tested the 170 samples and determined that 55 of them were positive samples. We cultured 55 samples according to the hospital's test, and finally obtained only 34 dengue viruses and conducted subsequent studies. 

2. Lines 55-59: “DENV, as a member of the genus Flavivirus in the family Flaviviridae, is an enveloped, single stranded, positive-sense RNA virus [8]. The DENV genome is approximately 11,000 nucleotides in length and encodes three structural proteins, namely, the capsid (C), premembrane (prM), and envelope (E) proteins, and seven nonstructural (NS) proteins (NS1,NS2A, NS2B, NS3, NS4A, NS4B and NS5) in a single open reading frame (ORF) [9,10].”References 8, 9 and 10 are not appropriate for the citation here.

Answer: References 8, 9 and 10 has been modified to new 8 “Characterization of dengue virus resistance to brequinar in cell culture”. The changed part is in the 61 lines of the manuscript.61

3. Lines 60-63: “There are five DENV serotypes…classified into different subtypes [11]”? Five DENV serotypes are controversial due to lack of evidence in humans. Major literature states 4 DENV serotypes. There is no citation of 5 serotypes and Reference 11 is inappropriate here. Different subtypes?

Answer: The five serotypes have not been widely accepted and are only mentioned in a few articles, so we changed it to four serotypes. Different serotypes can be further differentiated into different genotypes. Reference has been changed to 9 “Niu C, Huang Y, Wang M, Huang D, Li J, Huang S, et al. Differences in the Transmission of Dengue Fever by Different Serotypes of Dengue Virus. Vector Borne Zoonotic Dis. 2019. doi: 10.1089/vbz.2019.2477. PubMed PMID: 31503521.” The changed part is in the 63-64 lines of the manuscript.

4. Line 83: “Guangdong Province sees a continuous increase in the number of patients”?

Answer: This is not clearly described in the manuscript. We have corrected the expression in the manuscript. As a result of rapid economic growth, thriving tourism, and the greenhouse effect, there were many patients infected with dengue fever in Guangdong Province, and a serious outbreak occurred in 2014. The changed part is in the 83-85 lines of the manuscript. 

5. Lines 69 to 77: They provided a review of DENV epidemiology in Guangdong up to 2014.What happened between 2015 and 2017 before their study in 2018?

Answer: Guangdong Province is a coastal city in southern China and a city with a large population. The dengue virus is imported into Guangdong Province every year to cause an outbreak. In 2014, a serious outbreak broke out in Guangdong Province, and then there will be a small outbreak every year from 2015 to 2018. The following is an article about the report of Dengue cases in Guangdong Province from 2014 to 2018. For details, please see Figure 2 in the article. (Article name: Spatiotemporal analysis of the dengue outbreak in Guangdong Province, China). It can be seen from the figure that there are cases reported in Guangdong every year in the years after 2014, and the number of cases in 2016-2018 shows an increasing trend. The following figure is taken from the above paper.

6. Lines 99-100: “IgM and IgG enzyme-linked immunosorbent assay (ELISA) kits were used to confirm dengue infection”. Please describe which commercial kits were used or in-house assays.

Answer: The test kit used in hospitals is the Diagnostic Kit for Dengue Virus NS1 Antigen (ELISA) and the Dengue IgG/IgM Combo Test Card. The kit information is supplemented in the supplemental material.

Product Name Supplier

Dengue IgG/IgM Combo Test Card Xiamen Boson Biotech Co ,Ltd

Diagnostic Kit for Dengue Virus NS1 Antigen (ELISA) BEIJING WANTAI DRD CO., LTD

7. Lines 100-101: “A total of 170 serum samples were available”. These are likely acute samples. Which days post onset of symptoms were the samples collected?

Answer: Our samples were taken from the hospital and the hospital sampled suspected dengue patients on the day they visited the clinic. All patients had fever > 37.5 °C for less than 72 h.We have added the description to the article. The changed part is in the 99-100 lines of the manuscript.

8. Lines 112-113: “The amplified PCR products…” What are the PCR conditions? Reference 24 was real-time RT-PCR and not relevant here.

Answer: We may don't have a clear statement in the manuscript. RNA was reverse-transcribed into cDNA. Then DENV serotyping was carried out by multiplex RT-PCR. The changed part is in the 114-115 lines of the manuscript.

9. Line 117: “DENV-2 standard Hawaii strains”?

Answer: This is not clearly described in the manuscript. it is written like DENV-1. Now DNVE-2 strains GenBank accession number has been modified to KM279569. The changed part is in the 118-119 lines of the manuscript.

10. Table 1 “Primer sequence, and size of RT-PCR product…” There is no information of the size of RT-PCR product. At least, the genome positions of primers should be presented. There is no reference of these.

Answer: Size of RT-PCR product and genome positions of primers were added to table 1. references are labeled in the method.

11. Table 2: The genome positions of primers should be presented.

Answer: genome positions of primers were added to table 2.

12. Lines 150-152: “…we set up a global dataset that involved all samples and DENV-1 and DENV-2 sequences available from the GenBank…”? How many sequences from the GenBank were selected in the analysis and what are selection criteria? Do they include representative sequences from different genotypes within each serotype?

Answer: 3,529 sequences of DENV-1 were used for the study, and 3,542 sequences of DENV-2 were used for the study. Our dataset downloads all DENV-1 and DENV-2 sequences from genbank and screens the downloaded sequences (removing sequences with incomplete information such as lack of acquisition time and location, missing E gene sequences). The sequence of the data set is representative.

13. Lines 201-202: “… the entropy-based index presented by Xia. The ….Iss

Answer: The DAMBE software is used to calculate the base substitution saturation of the sequence. This calculation program was designed by the designer (Xia) of the DAMBE software. The changed part is in the 210-211 lines of the manuscript.

14. Lines 336-338: “Since Aedes albopictus is the main medium for DENV transmission…eggs and offspring through winter [34]”. Aedes lbopictus is the main medium for DENV? Reference 34 is about program interface and is an irrelevant citation.

Answer: Aedes albopictus is the main medium for DENV; The reference is not appropriate and has been changed to new article -“Guo X, Zhao T, Dong Y, Lu B. Survival and replication of dengue-2 virus in diapausing eggs of Aedes albopictus (Diptera: Culicidae). J Med Entomol. 2007;44(3):492-7. doi: 10.1603/0022-2585(2007)44[492:sarodv]2.0.co;2. PubMed PMID: 17547236.” The changed part is in the 363 lines of the manuscript.

15. Lines 332-333: “Others were localized strains after vertical transmission”. Vertical transmission has been reported to be very insufficient. Is vertical transmission the only mechanism to explain localized strains? How about silent transmission or sporadic cases plus under-reporting?

Answer: It is undeniable that the sequence does have a great impact on the experimental results. The data set used in this study consisted of downloading all dengue virus data from genbank and screening out the invalid sequences. The data set contained a large number of sequences to study. For traceability we also used blast to find kinship sequences, and the results presented a continuous cross-year epidemic. The coverage of the sequence has a great impact on the study, such as silent transmission or sporadic cases plus insufficient reporting, which may lead to false inter-annual transmission. We have already described this in the discussion of the manuscript. The changed part is in the 385-388 lines of the manuscript.

Reviewer #2

1. DENV-1 is sub-divided into five genotypes: I, II III, IV and V. DENV-2 is classified as six genotypes: Asian I, Asian II, Asian/American, Cosmopolitan, American and sylvatic. These genotypes should be assigned in Figure 2 and specify the genotypes the presented cluster I-V belongs to.

Answer: We have modified the map to mark the genotype. Thank you very much for your suggestion

2. In Table 3, do location columns really indicate the place of infection? Is there any questioner asking travel history?

Answer: Yes, we asked about the travel situation of 34 patients. They did not travel before suffering from dengue fever. 

3. Many Asian I viruses were deposited to NCBI from China. Before reached to the conclusion that the origin of the virus isolated in this study derived from Southeast Asian countries, the author should pay more attention to the pre-existing Chinese viruses in phylogenetic analysis. Can you deny the possibility from Yunnan or other part of China?

Answer: The data set used to construct the maximum likelihood tree in this study was downloaded from NCBI and can be used for all DENV-1 and DENV-2 sequences analyzed, including Chinese sequences, and we screened these large numbers of sequences. Finally, the database used in the study was formed. The data set contains Chinese sequences such as Yunnan Province, Fujian Province, and Taiwan Province. The results in such a data set show that the sample sequence is not clustered with other Chinese sequences in a cluster. So the results of the ancestral position did not show that they were from other provinces in China. 

4. The sequences used in molecular clock tests were deviated to recent sequences.

Answer: Molecular clock detection is based on sequences in the data set. As we mentioned in the manuscript, the dataset we used to build the MCC tree will be detected by the molecular clock before the tree is reconstructed. The sequences contained in these data sets were obtained after BLAST, and they were sampled at different years, but most of them were sequences in recent years, so the results of molecular clock detection showed that they were biased toward the recent sequences. So we also used a uncorrelated lognormal relaxed clock when building the MCC tree.

5. Did the author calculate the Bayes factor for phylogeographic reconstruction in Figure 3?What was the cut-off value?

Answer: Yes, we counted it. Bayes factor cut-off is 3.

6. In line 173, why the ULRC was selected?

Answer: Uncorrelated lognormal relaxed clock (ULRC) is a molecular clock model used by BEAST software to calculate phylogenetic trees. In the study, we examined the molecular clock for the data set. The structure shows that the data set does not exhibit a strict molecular clock shape, so we chose this model.

7. In line 174, why the Bayesian skyline coalescent model was selected?

Answer: The Bayesian skyline coalescent model was used to construct the Bayesian skyline plot of the dataset. We used this model because we wanted to explore the population dynamics of the dataset. This mo model is not used to build the MCC tree, so it does not affect our construction of the MCC tree.

---

## [Decision Letter · Decision Letter 1]

21 Oct 2019

Molecular epidemiological characteristics of dengue virus carried by 34 patients in Guangzhou in 2018

PONE-D-19-19136R1

Dear Dr. Li,

We are pleased to inform you that your manuscript has been judged scientifically suitable for publication and will be formally accepted for publication once it complies with all outstanding technical requirements.

With kind regards,

Jason Blackard, PhD

Academic Editor

PLOS ONE

Additional Editor Comments (optional):

None

Reviewers' comments:

None

Reviewer's Responses to Questions

**Comments to the Author**

1. If the authors have adequately addressed your comments raised in a previous round of review and you feel that this manuscript is now acceptable for publication, you may indicate that here to bypass the “Comments to the Author” section, enter your conflict of interest statement in the “Confidential to Editor” section, and submit your "Accept" recommendation.

Reviewer #2: All comments have been addressed

2. Is the manuscript technically sound, and do the data support the conclusions?

Reviewer #2: Yes

3. Has the statistical analysis been performed appropriately and rigorously? 

Reviewer #2: Yes

4. Have the authors made all data underlying the findings in their manuscript fully available?

Reviewer #2: Yes

5. Is the manuscript presented in an intelligible fashion and written in standard English?

Reviewer #2: Yes

6. Review Comments to the Author

Reviewer #2: (No Response)

7. PLOS authors have the option to publish the peer review history of their article (what does this mean?). If published, this will include your full peer review and any attached files.

Reviewer #2: No

---

## [Editor Report · Acceptance letter]

25 Oct 2019

PONE-D-19-19136R1 

Molecular epidemiological characteristics of dengue virus carried by 34 patients in Guangzhou in 2018 

Dear Dr. Li:

I am pleased to inform you that your manuscript has been deemed suitable for publication in PLOS ONE. Congratulations! Your manuscript is now with our production department. 

With kind regards,

on behalf of

Dr. Jason Blackard 

Academic Editor

PLOS ONE